# Development of a Ratiometric Fluorescent Glucose Sensor Using an Oxygen-Sensing Membrane Immobilized with Glucose Oxidase for the Detection of Glucose in Tears

**DOI:** 10.3390/bios10080086

**Published:** 2020-07-29

**Authors:** Hong Dinh Duong, Ok-Jae Sohn, Jong Il Rhee

**Affiliations:** 1School of Chemical Engineering and Research Center for Biophotonics, Chonnam National University, Gwangju 61186, Korea; zink1735@gmail.com; 2JoinTree Co., GwangJu 62421, Korea; ojsohn8276@gmail.com

**Keywords:** glucose-sensing membrane, glucose oxidase, ratiometric fluorescent sensor, fluorescence quenching, tear glucose

## Abstract

Glucose concentration is an important parameter in biomedicine since glucose is involved in many metabolic pathways in organisms. Many methods for glucose detection have been developed for use in various applications, particularly in the field of healthcare in diabetics. In this study, ratiometric fluorescent glucose-sensing membranes were fabricated based on the oxygen levels consumed in the glucose oxidation reaction under the catalysis of glucose oxidase (GOD). The oxygen concentration was measured through the fluorescence quenching effect of an oxygen-sensitive fluorescent dye like platinum meso-tetra (pentafluorophenyl) porphyrin (PtP) by oxygen molecules. Coumarin 6 (C6) was used as a reference dye in the ratiometric fluorescence measurements. The glucose-sensing membrane consisted of two layers: The first layer was the oxygen-sensing membrane containing polystyrene particles (PS) doped with PtP and C6 (e.g., PS@C6^PtP) in a sol–gel matrix of aminopropyltrimethoxysilane and glycidoxypropyltrimethoxysilane (GA). The second layer was made by immobilizing GOD onto one of three supporting polymers over the first layer. These glucose-sensing membranes were characterized in terms of their response, reversibility, interferences, and stability. They showed a wide detection range to glucose concentration in the range of 0.1 to 10 mM, but high sensitivity with a linear detection range of 0.1 to 2 mM glucose. This stable and sensitive ratiometric fluorescent glucose biosensor provides a reliable way to determine low glucose concentrations in blood serum by measuring tear glucose.

## 1. Introduction

The determination of glucose concentration is of great interest in healthcare for diabetics, who account for about 5% of the world’s population [1]. The disease stems from insulin deficiency and metabolic disorders caused by hyperglycemia, which is characterized by blood sugar levels above or below the normal range of 80–120 mg/dL (4.4–6.6 mM) [2]. Therefore, many glucose sensors have been developed using several approaches, many of which have involved electrochemical and optical methods. 

Glucose biosensors using electrochemical and optical analysis technology occupy about 85% of the total biosensor market [2]. Each technology has pros and cons and needs to be adjusted to achieve more accurate results, reliable monitoring, and safety. Measurements by glucose biosensors can be separated to noninvasive and invasive methods; non-invasive methods are preferred due to their selectivity, reversibility, long lifetime, and fast and predictable response to changing glucose concentrations, which make them inexpensive on a large scale [3]. 

Among noninvasive optical glucose biosensors, fluorescence-based glucose biosensors have recently attracted interest because of the advantages of fluorescent molecules for biosensing, such as extreme sensitivity due to selective chemical reactions between functional groups of fluorescent dyes and analytes as well as high fluorescent quantum yield of dye even at trace level of samples, the need for less reagents, no damage to the host system, various signal outcomes, and their ability to display the structure and the distribution of biomolecules [4]. Fluorescence was first used to detect glucose in 1984 by S. Mansouri et al. [5], where Concanavalin A (ConA) was immobilized on a high molecular weight fluorescein-labeled dextran inside of a microdialysis fiber. The fluorescein-labeled dextran was displaced from the binding site by glucose, causing an increase in fluorescence in proportion to the glucose concentration. This mechanism is developed by immobilizing ConA on hydrogel sphere particles to increase the linear response up to 600 mg/dL [6]. In fact, the mechanisms of many glucose biosensors are based on changes in the concentrations of the components involved in the oxidation reaction of glucose during the catalysis of glucose oxidase (GOD). Oxygen consumption or hydrogen peroxide production is used in the fabrication of glucose sensors [7,8]. Fluorescence is also combined with the enzyme GOD to change the fluorescence state through energy transfer between the flavin group (FAD) of GOD and fluorescein-5(6)-carboxamidocaproic acid n-succinimidyl ester during glucose measurement [9]. In addition, the concept of glucose detection based on oxygen consumption in the glucose oxidation reaction under the catalysis of GOD is exploited since many fluorescent oxygen sensors are recently developed [10,11,12,13,14,15,16,17]. However, there is a need for a less invasive method to self-monitor glucose levels for diabetes patients [18]. One noninvasive method for determining the concentration of glucose in blood serum is to measure the concentration of glucose in tear fluid. The approach of using tear glucose is due to requests for sensitivity, ease of use, low cost, and minimal sampling and measurement steps of glucose sensors. The tear glucose of diabetics has been studied since the beginning of the last century [19] and in a recent contribution of Wang et al. [20], but the correlation between tear glucose levels and blood glucose levels in normal and diabetic patients remains controversial. Regardless of whether or not scientists agree with this concept, many glucose sensors have been developed that use tear glucose instead of blood glucose [21,22,23,24].

Among various fluorescent dyes and metalloporphyrins available for such purposes, platinum meso-tetra (pentafluorophenyl) porphyrin (PtP) is one of the most often used oxygen sensing probes [13,25]. The four electron-withdrawing pentafluorophenyl substituents of the PtP raise its redox potentials and electron density, which makes PtP more photostable and less sensitive to temperature than ruthenium complex (Ru(dpp)_3_) [10]. In addition, PtP is highly photostable in polymers such as polystyrene, thus making it well suited for the long-term continuous monitoring of oxygen [26,27]. 

Polystyrene (PS) has good optical properties along with acceptable permeability and solubility coefficients for oxygen. Its oxygen permeability is 0.88, and its quenching constant is typically smaller than that of silicone by a factor of 10–100 [10]. Therefore, PS is preferred as a supporting material in the fabrication of oxygen sensors based on PtP to prevent dye leakage [26,27,28,29].

Sol–gel of organically modified silicates has high oxygen permeability due to the high fractions of alkyl and aryl siloxanes contained within, which favor oxygen permeability and probe solubility. Therefore, sol–gels are often used to make oxygen sensors [30,31,32]. Particularly, sol–gel GA has shown good chemical stability and superior optical transparency in our previous studies. 3-glycidoxypropyltrimethoxysilane (GPTMS) contains an epoxy group that can be covalently bound to the amine groups of others during a gel process, while 3-aminopropyltrimethoxysilane (APTMS) has been widely used to introduce amino groups to bind various probes and biomolecules [33].

Ethyl cellulose (EC) is a derivative of cellulose in which most of the hydroxyl groups on the repeating glucose units are ethylated. EC has been widely used in matrices for hosting oxygen-sensing probes [34,35] due to its excellent oxygen permeability, high optical transparency, high mechanical strength, and good photo/thermal stability [10]. 

Polyurethanes (such as D4) form another large class of hydrogels. Hydrophilic supports such as D4 are characterized by a large number of hydrogen-bridging functions as hydroxyl, amine, or carboxamide groups linked to the polymer backbone. They are easily penetrated by aqueous solutions, but are rarely used in oxygen sensors due to the lack of permeation selectivity and slow diffusion of oxygen. However, these materials have excellent biocompatibility, and various enzymatic biosensors using polyurethanes, where both the enzyme and oxygen-sensing probes are immobilized into the polyurethane, have been reported [36]. Polyurethane is also used as a paint containing hydrophilic oxygen sensor particles [37,38].

In this work, the ratiometric fluorescent glucose-sensing membranes were developed from the ratiometric fluorescent oxygen-sensing membrane immobilized glucose oxidase (GOD) by using different supporting polymers. The glucose-sensing membranes were fabricated with two layers; the first layer was the oxygen-sensing membrane containing polystyrene particles (PS) doped with platinum meso-tetra (pentafluorophenyl) porphyrin (PtP) and coumarin 6 (C6) (e.g., PS@C6^PtP) in a sol–gel matrix of GA. The second layer was the immobilization of GOD onto three kinds of supporting polymers: sol–gel GA, polyurethane, and ethylcellulose (Scheme 1). The working mechanism of the first layer (fluorescence transducer) was based on the fluorescence quenching effect of PtP by oxygen molecules and C6 as a reference dye for use in ratiometric fluorescence measurements. The change in oxygen concentration from the oxygen consumption of the glucose oxidation reaction on the second layer under the catalysis of GOD was reflected in the changes in the fluorescence intensity of PtP and its proportion to the glucose concentration. The properties of the glucose-sensing membranes were investigated ratiometrically, and their response to glucose was also used to help determine glucose concentrations in tears. 

## 2. Materials and Methods

### 2.1. Materials

Glucose, glucose oxidase (GOD, 26,820 unit(U)/g-solid, from *Aspergillus niger*), coumarin 6 (C6), 3-aminopropyltrimethoxysilane (APTMS), 3-Glycidoxypropyltrimethoxysilane (GPTMS), bovine serum albumin (BSA), iron chloride, poly(vinyl pyrrolidone) (PVP, M.W. ≈ 55,000), styrene, 2,2’-azobisisobutyronitrile (AIBN), sodium dodecyl sulfate (SDS), L-ascorbic acid, uric acid, and acetaminophen were purchased from Sigma-Aldrich Chemical Co. (Seoul, Korea). Pt(II) meso-tetra (pentafluophenyl) porphine (PtP) was obtained from Frontier Scientific Co. (Logan, UT, USA). Other chemicals such as ethanol, tetrahydrofuran (THF), hydrochloric acid, sodium hydroxide, sodium phosphate, sodium chloride, and sodium bicarbonate were of analytical grade and used without further purification. 

### 2.2. Preparation of Dye-Doped Polystyrene (PS) Particles

In a typical synthesis, 25 mL of ethanol and 5 mL of deionized water and 0.25 g of poly (vinyl pyrrolidone) (PVP) were placed in a three-neck flask (100 mL) equipped with a condenser. This solution was heated at 80 °C for 30 min, then 2.5 mL of styrene and 100 µL of AIBN were sequentially added to the solution. Next, the polymerization was allowed to proceed for 6 h at 80 °C, with magnetic stirring applied throughout the entire synthesis. Finally, the suspension of PS particles was cooled to room temperature. The monodispersed PS particles were collected via centrifugation at 5000 rpm for 15 min, then washed with ethanol three times before being dried at room temperature.

At this point, 200 mg of PS particles was added to 10 mL of 2% SDS solution, which was then stirred for about 4 h to achieve homogeneous dispersion of particles. Afterward, 1 mL of 0.2 mM C6 in THF was added dropwise to the solution of PS particles under strong stirring for 12 h. The PS particles doped with C6 (PS@C6) were collected via centrifugation at 5000 rpm for 15 min, then washed with distilled water three times and dried at room temperature. The second step for loading PtP on PS@C6 was similar to the procedure for dyeing C6 in PS particles. First, only 2.5 mL of 10 mM PtP in THF was added dropwise to the solution of PS@C6 particles in 10 mL of SDS solution. The next step was stirring for 12 h, collecting the PS particles doped with C6 and PtP (PS@C6^PtP) by centrifugation, and finally washing with distilled water several times and drying at room temperature.

### 2.3. Preparation of the PS@C6^PtP Membrane

Sol–gel GA was formed by the hydrolyzation and polymerization of a mixture of APTMS and GPTMS in ethanol at a volumetric ratio of 6.5%:25%:68.5%, respectively, and the volume of HCl (37%) was 4% (v/v) of the sol–gel GA volume. After the addition of HCl, the sol–gel GA was kept at room temperature for 4 h before being used in the subsequent steps. 

Next, 5 mg of dye-doped PS particles (PS@C6^PtP particles) was mixed with 1 mL of the sol–gel GA and shaken for 2 h before being coated on the bottom of a well in a 96-well microtiter plate (10 µL/well) and then finally dried at 60 °C for 12 h.

### 2.4. Immobilization of GOD onto the PS@C6^PtP Membrane

For the immobilization of GOD, there were three kinds of polymers: 10 wt% ethyl cellulose (EC) in ethanol, 10 wt% polyurethane hydrogel (D4) in ethanol and water (9:1 in v/v%), and sol–gel GA. A layer of a given amount of GOD in 20 µL EC or D4 or 10 µL GA was coated over the surface of the PS@C6^PtP membrane of each well in a 96-well microtiter plate. 

For the immobilization of GOD, 10 U, 20 U, 40 U, 50 U, 60 U, and 100 U of GOD were tested to find the optimal amount. The immobilization efficiency of GOD onto the PS@C6^PtP membrane using three different supporting materials was calculated based on the Bradford method. The PS@C6^PtP membranes immobilized with various amounts of GOD (GOD = PS@C6^PtP) were measured with different glucose concentrations. The sensitivity of the GOD = PS@C6^PtP membranes was evaluated based on the slope value (SI), that is, the ratio of the fluorescence intensities at two emission wavelengths (*λ_em_* = 475 nm and *λ_em_* = 635 nm) with respect to glucose concentration to obtain the optimal amount of GOD for immobilization. The kinetic parameters, maximal reaction rate (V_max_), and Michaelis–Menten constant (K_m_) of the immobilized GOD were determined from the Lineweaver–Burk plot based on the ratio of the fluorescence intensities at *λ_em_* = 475 nm and *λ_em_* = 635 nm. 

### 2.5. Fluorescence Measurements

The response of the GOD = PS@C6^PtP membranes to different glucose concentrations in the range of 0.1 mM to 10 mM was determined using a multifunctional fluorescence microtiter plate reader (Safire^2^, Tecan Austria GmbH, Wien, Austria). Data were collected from the fluorescence intensities of the GOD = PS@C6^PtP membrane at two emission wavelengths (*λ_em_* = 475 nm and *λ_em_* = 635 nm) with an excitation wavelength of 400 nm (*λ_ex_* = 400 nm). 

The reversibility of the GOD = PS@C6^PtP membranes was examined with 2 mM glucose and distilled water. The glucose-sensing membrane was first exposed to distilled water and then to 2 mM glucose solution, and this process was repeated. The microplate reader was set for fluorescence measurements against time with an interval of 30 s during measurements. 

The effects of pH and temperature on glucose measurement were investigated next. The GOD = PS@C6^PtP membranes were tested at different temperatures (25, 30, 33, 35, and 37 °C) or in the range of pH 5.0 to pH 9.0 at glucose concentrations ranging from 0.1 mM to 10 mM. The long-term stability of the GOD = PS@C6^PtP membranes was evaluated with various glucose concentrations by measuring the initially obtained fluorescence intensity and that after one month of use. 

The interfering effects of some components contained in blood serum, such as ions (Na^+^, Cl^−^, HCO_3_^−^, and Fe^3+^) and albumin (BSA), on the glucose-sensing membranes were investigated. The glucose-sensing membranes were measured with 145 mM Na^+^, 106 mM Cl^−^, 30 mM HCO_3_^−^, 1.625 mg/L Fe^3+^, and 5 g/dL BSA at 1 mM glucose concentration. 

### 2.6. Ratiometric Fluorescence Method and Data Analysis

A ratiometric fluorescence method for the GOD = PS@C6^PtP membranes was based on the ratio of the fluorescence intensities at two emission wavelengths (*λ_em_* = 635 nm (FI_635_) and *λ_em_* = 475 nm (FI_475_)) as follows:R = FI_635_/FI_475_(1)

The differences in the fluorescence intensities and slopes of the linear ranges of the GOD = PS@C6^PtP membranes at different interferences were assessed through one-way analysis of variance (ANOVA). Significant differences between samples were accepted with *p*-value < 0.05. Statistical tests were performed using InStat software (vers.3.01, Graph Pad Software Inc, San Diego, CA, USA).

### 2.7. Applications

The glucose-sensing membranes (i.e., the GOD = PS@C6^PtP membranes) were investigated for the detection of glucose in the artificial tear. The artificial tear solution consisted of 10 mM phosphate saline buffer (PBS, pH 7.0), 10 µM uric acid, 100 µM ascorbic acid, and 10 µM acetaminophen. The results obtained from the artificial tear solutions with ratiometric fluorescence calculation were compared with those obtained from standard glucose solutions.

## 3. Results and Discussion

### 3.1. Mechanism of the Fluorescence Quenching Effect of the Oxygen-Sensitive Dye (PtP)

The operation mechanism of oxygen-sensitive fluorescent dyes such as PtP in the presence and absence of oxygen is illustrated in Scheme 2. In the absence of oxygen, when a fluorescence molecule is excited, its energy level changes from ground state to excited state. After a certain time, its energy returns to the ground state and during that time it emits fluorescence (photons). In the presence of oxygen, when a fluorescence molecule is in the excited state and oxygen is in the ground state, a collision occurs between two molecules, which leads to an energy transfer between them, resulting in decreases in the fluorescence intensity and the lifetime of the fluorescence molecule as well as a transformation of oxygen from its ground state (triplet ^3^O_2_) to its excited state (singlet ^1^O_2_).

### 3.2. Properties of the PS@C6^PtP Membrane

As shown in Figure 1a (left), the synthesized polystyrene particles (PS) are spherical and have a diameter of about 1 µm. Fluorescent dyes (C6, PtP) dissolved in hydrophobic solvent could be fully loaded on PS particles by exploiting the properties of PS such as swelling and shrinking in hydrophobic and hydrophilic solvents, respectively, as shown in Figure 1a (middle) for C6 dye and in Figure 1a (right) for both C6 and PtP dyes. The PS particles containing both C6 and PtP dyes are used to make an oxygen-sensing membrane (PS@C6^PtP membrane). The PS@C6^PtP membrane shows emission band edges at *λ_em_* = 635 nm for PtP and at *λ_em_* = 475 nm for C6. This is shifted about 10 nm after loading the PtP dye on the PS particles (PtP dye: *λ_em_* = 645 − 650 nm). The other properties of the PtP dye in the PS@C6^PtP membrane are stable for sensing oxygen.

The morphology of the PS@C6^PtP membrane is presented in Figure 2a. The AFM data indicate that the sol–gel GA membrane is a thin and smooth membrane with a surface mean roughness (Ra) of 3.558 nm and a root mean square roughness (Rq) of 4.335 nm (Figure 2a, left). By contrast, the PS@C6^PtP membrane using the sol–gel GA as a supporting material has a surface mean roughness (Ra) of 4.991 nm and a root mean square roughness (Rq) of 6.315 nm (Figure 2a, right). Thus, the surface morphology of the PS@C6^PtP membrane is much rougher than that of the sol–gel GA membrane. SEM images have also confirmed the surface morphologies of these membranes.

As shown in our previous study [33,39], sol–gel is a good material for the penetration and the convection of oxygen. In this work, the PS@C6^PtP membrane shows high sensitivity to different oxygen concentrations (Figure 2b) along with excellent reversibility in the presence and the absence of oxygen, since Relative standard deviation (RSD) is 2.64% and 3.85% at 0% and 100% oxygen, respectively. In the PS@C6^PtP membrane containing both C6 and PtP fluorescent dyes, since the fluorescent emission of the PtP dye is quenched at high oxygen concentration, the fluorescent emission of the C6 dye can be clearly observed. As shown in Figure 2c, the orange color of the C6^PtP dye in the PS@C6^PtP membrane is clearly seen at low oxygen concentrations, particularly in the absence of oxygen. By contrast, the green color of the C6 dye appeared with increasing oxygen concentration and could clearly be seen at 100% oxygen. The color change of the PS@C6^PtP membrane that occurs when it is exposed to different oxygen concentrations can quickly be recognized within several seconds. 

### 3.3. Properties of the GOD-Immobilized PS@C6^PtP Membrane (GOD = PS@C6^PtP Membrane)

Based on SEM images, the PS@C6^PtP membranes immobilized with GOD on three supporting materials look substantially different from each other (Figure 3). Due to the simple capture of GOD into the polymer matrix of EC, the second layer of the PS@C6^PtP membrane looks like rolls because of the waves on the surface of the glucose-sensing membrane (Figure 3a). As GOD immobilized on D4 hydrogel is somewhat visible on the surface of the sensing membrane (Figure 3b), a large number of functional groups of D4 polymer indicates that D4 can serve as a good supporting material for GOD. 

Figure 3c shows the best image for using the sol–gel GA as a supporting material for GOD immobilization, where it can be seen that the formation of sol–gel GA clearly arranged clusters of GOD on the sensing membrane when added to the aqueous solution of GOD. This arrangement may be attributable to the typical covalent binding between GA and GOD.

According to data collected from the Bradford protein assay, the results of which are shown in Figure 4 (below), the immobilization efficiencies of GOD on different supporting polymers vary substantially. D4 polymer appears to be the best supporting material for the immobilization of GOD in most of the enzymes used. The covalent binding between isocyanate of D4 with the amine group of GOD can occur as shown in Scheme 3a, leading to tight bonds of GOD with its supporting material. However, the binding ability of sol–gel GA with GOD is better than that of D4 polymer with GOD. This is because GA can be combined with GOD through the two methods shown in reaction Scheme 3b,c. That is, the epoxy group of GPTMS and the amine group of APTMS can covalently bind with the amine group and the carboxyl group of GOD, respectively. The immobilization efficiency of GOD on sol–gel GA was in fact not higher than those using EC and D4 as supporting materials. This resulted in a thin GA membrane due to the low viscosity of sol–gel GA, which is not spatially sufficient to capture GOD in its matrix. Meanwhile, GOD immobilized into EC is simply a capture of GOD into the EC matrix, but high immobilization efficiency of GOD is observed in most of the GOD used, and it seems to be better than that of the sol–gel GA. Therefore, even though the immobilization capacities of the supporting materials (EC, D4, and GA) for GOD are different, an amount of around 40–60 U GOD can be chosen to fabricate the oxygen-sensing membrane in further immobilization experiments. 

As shown in Figure 4 (upper), the PS@C6^PtP membranes that used EC, D4, and GA as supporting materials for the immobilization of GOD were highly sensitive to glucose in the linear range of 0.1–2 mM across all amounts of GOD used. The sensitivity of the GOD = PS@C6^PtP membranes increased with increasing amounts of GOD used (i.e., increases in the slope value). However, using 100 U GOD did not yield better results when using EC as a supporting material (SI_100U_ = 0.983); this may have been due to an overload of GOD in the EC matrix. 

With using D4 and GA as supporting materials, the use of 100 U GOD leads to the best results with SI values of 0.1436 and 0.1163, respectively. Meanwhile, the sensitivity of the GOD = PS@C6^PtP membrane shows excellent results (SI~0.122) with smaller amounts of GOD (40–60 U) when using D4 and GA, while using 10–20 U GOD still leads to a high response of the GOD = PS@C6^PtP membranes to all glucose concentrations. 

As was theoretically predicted, too much immobilized enzyme can narrow the detection range or prevent the transport of analyte from contacting the transducer membrane. However, too little immobilized enzyme results in slow reaction and response as well as low sensitivity of the sensing membrane or instability over a long period of use. Therefore, it is confirmed that 50 U GOD is a suitable amount for the immobilization of GOD on three supporting materials (EC, D4, and GA) in order to compare the response of the GOD = PS@C6^PtP membranes in the next measurements. 

In addition, the slope values in Figure 4 (upper) somewhat indicated the tight chemical coupling of GOD to supporting materials such as D4 and GA which resulted in higher sensitivity of the GOD = PS@C6^PtP membranes, as compared with the case of using EC along with small amounts of GOD. However, the surface structure of the supporting material could be swollen, contracted, or modified along the time of use, leading to a decrease or an increase in the sensitivity of the sensing membranes. Moreover, the response of the glucose-sensing membranes depends on the stability of the binding of GOD to its supporting matrices for long-term use. Therefore, the data shown in Figure 4 indicate that while they should be preferred at the beginning of experiments, using D4 and GA is not better than using EC for GOD immobilization. 

### 3.4. Response of the Glucose-Sensing Membranes (GOD = PS@C6^PtP Membranes)

As shown in Figure 5, the emission band edges of the GOD = PS@C6^PtP membranes are *λ_em_* = 635 nm for PtP and *λ_em_* = 475 nm for C6. Therefore, the change in fluorescence intensities of the GOD = PS@C6^PtP membrane with respect to glucose concentrations is recognized at the wavelength of *λ_em_* = 635 nm, whereas *λ_em_* = 475 nm is used as a reference emission wavelength.

When using EC as a supporting material for the immobilization of GOD, the consumption of oxygen in the oxidation reaction of glucose led to an increase in the fluorescence intensity of the GOD = PS@C6^PtP membrane (Figure 5a). 

The fluorescence intensity of the GOD = PS@C6^PtP membrane increased with increasing glucose concentrations in the range of 0.1–10 mM. Its linear detection range was 0.1–2 mM with high regression coefficient value of *r*^2^ = 0.994 and a limit of detection (LOD, S/N = 3) of 0.025 mM. The activity of GOD immobilized onto the EC matrix was evaluated via Michaelis–Menten kinetics. The kinetic parameters were calculated from the ratio of two emission fluorescence intensities at *λ_em_* = 635 nm and *λ_em_* = 475 nm. A maximal reaction rate (V_max_) of 476.2 mM/min and Michaelis–Menten constant (K_m_) of 0.286 mM were obtained from the Lineweaver–Burk plot. 

When using D4 as a supporting material for GOD immobilization (Figure 5b), the response of the GOD = PS@C6^PtP membrane was similar to that of using EC. The fluorescence intensity of the GOD = PS@C6^PtP membrane increased with increasing glucose concentrations in the range of 0.1–10 mM, and its linear detection range was 0.1–2 mM with an LOD of 0.029 mM. The kinetic parameters of the maximal reaction rate (V_max_) and the Michaelis–Menten constant (K_m_) were 140.8 mM/min and 0.366 mM, respectively. 

When using sol–gel GA as a supporting material for immobilization of GOD (Figure 5c), the response of the GOD = PS@C6^PtP membrane was similar to those of the other two cases (EC and D4). The fluorescence intensity of the GOD = PS@C6^PtP membrane increased with increasing glucose concentrations in the range of 0.1–10 mM, and its linear detection range was 0.1–2 mM with LOD of 0.043 mM. The kinetic parameters, such as the maximal reaction rate (V_max_) and the Michaelis–Menten constant (K_m_), were calculated to be 73 mM/min and 0.364 mM, respectively.

Thus, according to data collected from the kinetics of GOD immobilized onto different polymers (V_max_ and K_m_), EC is shown to be a good supporting polymer for the transport of substrate in the oxidation reaction of glucose, compared to D4 and GA, even though its immobilization capability for GOD is limited, resulting in a smaller K_m_ value than the other two cases. However, the GOD = PS@C6^PtP using GA as supporting material for GOD immobilization shows better response in the high glucose concentration range (2–10 mM) than the other two cases (EC and D4), indicating that it is suitable for glucose measurements at both low and high glucose concentrations.

In addition, based on the photo images of the response of the PS@C6^PtP membrane shown in Figure 2c and a comparison with the graphs on the left of Figure 5, the oxygen concentration in the detection of glucose could be varied in the range of 15–21%. 

Generally, each glucose sensor exhibits its pros and cons in terms of detection range, accuracy, selectivity, repeatability, and stability because conditions of sensor fabrication are different [40,41,42,43,44]. However, in comparison with other sensors for glucose detection in tear fluid, such as the research of La Belle et al. using electrodes for detection of glucose in tear fluid in the range of 0–1 mM [21], or the research of Wang et al. with tear glucose detection to 2.2 mM [20], results of the ratiometric fluorescent glucose sensors in this work are somewhat better.

### 3.5. The Reversibility of the Glucose-Sensing Membranes (GOD = PS@C6^PtP Membranes)

The PS@C6^PtP membrane showed very high sensitivity and reversibility when exposed to a repeated cycle of low and high concentrations of oxygen (Figure 2b). When GOD was immobilized onto the PS@C6^PtP membrane, the reversibility of the glucose-sensing membrane was still excellent when exposed to a repeated cycle of glucose concentrations of 0 and 2 mM (Figure 6). 

All the GOD = PS@C6^PtP membranes with different supporting polymers showed fast recovery between 0 and 2 mM glucose with small values of RSD: 0.32% at 0 mM and 0.15% at 2 mM glucose for EC, 0.38% at 0 mM and 0.79% at 2 mM for D4, and 23% and 0.33% at 0 and 2 mM, respectively, for GA as a supporting material.

These results also indicated that the thickness of the second layer containing GOD did not affect the contact between the PS@C6^PtP membrane and oxygen. Among the three kinds of supporting materials used for the immobilization of GOD, sol–gel GA seems to be the best for the high reversibility of the PS@C6^PtP membrane, since the saturation point is reached in the shortest time when using sol–gel GA. The excellent response of the GOD = PS@C6^PtP membrane could be attributed to the thin and uniform sol–gel GA membrane in the first and second layer. 

### 3.6. Effects of pH and Temperature on the GOD = PS@C6^PtP Membranes

For biosensors using enzymes, pH and temperature are important parameters affecting measurement results. Solution pH can increase or decrease the activity of the enzyme in a sensor, and consequently increase or decrease the efficiency of any catalytic reactions. 

As shown in Figure 7, GOD immobilized on/in EC and D4 matrix exhibited better sensing performance in the pH range of 5–7 to the pH range of 8–9. The ratio of the fluorescence intensities at *λ_em_* = 475 nm and *λ_em_* = 635 nm of the glucose-sensing membrane did not change significantly at any glucose concentrations in the range of 0.1–10 mM in the pH range of 5–7. 

Meanwhile, pHs from 7 to 9 are more favorable for GOD immobilized on/in GA supporting material than pH 5 to pH 6. The ratio of the fluorescence intensities of the glucose-sensing membrane (at *λ_em_* = 75 nm and *λ_em_* = 635 nm) did not change significantly at any glucose concentrations in the range of 0.1–10 mM in the pH range of 7–9. This is attributed to the use of sol–gel GA, which has amine functional groups which lead to the preference of GA in alkaline medium. Thus, the pH of the glucose solution should be pH 7, which is suitable for GOD = PS@C6^PtP membranes using EC, D4, and GA.

Like pH, temperature can speed up or slow down catalytic reactions. Figure 8 shows the response of the GOD = PS@C6^PtP membrane with different glucose concentrations at different temperatures. The glucose-sensing membranes using EC and D4 as supporting matrices do not appear to be substantially affected by temperature in the range of 25–37 °C at the glucose concentration range of 0.1–10 mM. 

The properties and the thickness of the EC and D4 layers can serve to prevent temperature effects on the operation of the glucose-sensing membranes in this temperature range. While the GOD = PS@C6^PtP membrane using GA shows low sensitivity of the sensing membrane at 37 °C, other temperatures in the range of 25–35 °C did not affect the sensitivity of the GOD = PS@C6^PtP membrane at any glucose concentrations. 

Sol–gel GA was identified as a good material for heat transfer in our previous study [45,46]. Moreover, by using a thin layer of GA, it is easy to make the GOD = PS@C6^PtP membrane affected by temperature. In fact, the higher the temperature is, the higher the activity of the enzyme is, but in this case, the sensitivity of the glucose-sensing membrane decreased at high temperatures. This may be attributed to the movement of PtP particles in the PS@C6^PtP membrane at high temperatures where the fluorescence emission overlaps when excited.

In order to maintain a long lifetime of GOD and high sensitivity and reversibility of the GOD = PS@C6^PtP membrane, 30 °C was chosen as a favorable temperature for the measurement in this work.

### 3.7. Selectivity and Long-Term Stability of the GOD = PS@C6^PtP Membranes

As described in previous sections, the GOD = PS@C6^PtP membranes show a detection range of 0.1–10 mM glucose. This concentration range is suitable for the detection of glucose in blood serum where glucose changes in the concentration range of 2.5–7.1 mM for nondiabetic people and outside this range for diabetics. Therefore, some compounds such as albumin, Cl^-^, HCO_3_^−^, Fe^3+^, and Na^+^ ions are typically present in blood serum. The normal concentrations of these components in blood serum are in the ranges of 96–106 mM/L for Cl^−^, 20–30 mM for HCO_3_^−^, 0.5–1.76 mg/L for Fe^3+^, 135–145 mM/L for Na^+^, and 2.9–5.5 g/dL for albumin. In this work, the GOD = PS@C6^PtP membranes using EC, D4, and sol–gel GA as supporting materials were measured at 1 mM glucose in the absence and the presence of 106 mM/L Cl^−^, 30 mM/L HCO_3_^−^, 1.625 mg/L Fe^3+^, 145 mM/L Na^+^, and 5 g/dL BSA.

If the ratio of the fluorescence intensities at *λ_em_* = 635 nm and *λ_em_* = 475 nm (FI_635_/FI_475_) of the GOD = PS@C6^PtP membranes using EC, D4, and GA at 1 mM glucose is set as 100%, the FI_635_/FI_475_ of the glucose-sensing membranes at 1 mM glucose, along with a given amount of interference, is expressed as a percentage based on the control sample (1 mM glucose alone). As shown in Figure 9, the presence of interferences at high levels of concentration threshold appears to have less of an effect on the glucose-sensing membranes in all cases (EC, D4, and GA), because the percentage deviation of samples containing interference is about 1.0–12% compared to the control sample. In addition, the results of statistical analysis show that the *p-*values are always larger than 0.05 (*p-*value > 0.05), when compared with the control sample with the samples containing the interferences. This means that there were no significant difference between samples and no significant influence of the factors as mentioned above on all the GOD = PS@C6^PtP membranes using EC, D4, and GA.

After one month of continuous measurements of the GOD = PS@C6^PtP membrane, the sensitivities of all glucose-sensing membranes were found to be quite good (Figure 10). The slope values (SI) of the linear calibration curves in the glucose concentration range of 0.1–2 mM did not change significantly in all cases. When using EC as a supporting material for the immobilization of GOD, the SI values were SI_initial_ = 0.0697 and SI_1month_ = 0.0623; for D4 they were SI_initial_ = 0.066 and SI_1month_ = 0.0622; for GA they were SI_initial_ = 0.0598 and SI_1month_ = 0.0545. The PS@C6^PtP membrane and the GOD = PS@C6^PtP membranes on different supporting materials (EC, D4, and GA) showed high stability, making them appropriate for long-term use.

### 3.8. Application of the GOD = PS@C6^PtP Membrane

According to the results of the GOD = PS@C6^PtP membrane for the detectability of low glucose concentrations found in this study, the GOD = PS@C6^PtP membrane was investigated for the detection of glucose in artificial tears. The response of the GOD = PS@C6^PtP membranes to standard glucose solutions was compared with that to artificial tear solutions. 

As shown in Figure 11, the presence of certain factors in the tears did not significantly affect the response of the GOD = PS@C6^PtP membranes to different glucose concentrations in the artificial tear solution. The slope values of the linear calibration curves in the glucose concentration range of 0.1–2 mM did not change significantly in any cases. When using EC as a supporting material for the immobilization of GOD, the SI values were SI_std glucose_ = 0.0555 and SI_tear glucose_ = 0.0503; for D4 they were SI_std glucose_ = 0.0821 and SI_tear glucose_ = 0.0846; for GA they were SI_std glucose_ = 0.0561 and SI_tear glucose_ = 0.057. The percentage deviation (100 × (R_std glucose_ − R_tear glucose_)/R_std glucose_) of ratiometric fluorescence intensities (R = FI_635_/FI_475_) in glucose concentrations between standard solution and artificial tears in the detection range of 0.1–10 mM was from −1.5% to 9.0% in all cases.

## 4. Conclusions

Glucose-sensing membranes were successfully developed by combining an oxygen-sensing membrane (i.e., the PS@C6^PtP membrane) with a layer of GOD immobilized onto different supporting materials. The GOD = PS@C6^PtP membranes showed high sensitivity to the glucose concentration range of 0.1–10 mM, and a particularly highly sensitive linear detection range from 0.1 mM to 2 mM glucose. The encapsulation of the oxygen-sensing dye (PtP) and the reference dye (C6) into polystyrene particles (PS) prevented the leakage of dyes from measurements and produced highly stable membranes to oxygen. The different supporting materials (EC, D4, and GA) used for GOD immobilization showed certain advantages and disadvantages that may be appropriate for different situations. The high sensitivity of the GOD = PS@C6^PtP membranes to low glucose concentrations can be used as another mode to evaluate glucose concentration in blood serum; that is, it can be used for glucose measurement in tears.

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
