# Peer review of "Development of a Ratiometric Fluorescent Glucose Sensor Using an Oxygen-Sensing Membrane Immobilized with Glucose Oxidase for the Detection of Glucose in Tears"

_biosensors, 2020, doi:10.3390/bios10080086_

Round 1

Reviewer 1 Report

This is an interesting work. This manuscript describes a work to fabricate a ratiometric fluorescent glucose sensor containing an oxygen-sensing membrane immobilized with glucose oxidase. It evaluates three different supporting materials and performance of this sensor, which has been utilized for glucose detection in tears. This work provides a reference for the development of glucose sensors which can be used in the clinical treatment of diabetes.

However, there are several concerns.

(1) In Figure 1 a, there is no scale bar mark, which should be marked in the picture. And the same problem appears in Figure 3.

(2) In Figure 2a, the result of left figure shows a rougher surface than that in the right figure. However, the statement in the main text leads to the opposite conclusion. Please explain.

(3) In the main text, the SI value of GA(100 U GOD) is stated as 0.1635, which might not be consistent with the corresponding data in Figure 4?

(4) In Figure 5, the layout of the pictures doesn’t look good because two pictures partially overlap. It might be better to put the two figures side by side.

(5) There is a mistake on line 357, Fig. 1c might be changed to Fig. 1b?

(6) In the experiment of evaluating the long-term stability of the GOD=PS@C6^PtP membranes, there are only two time points settled, which might be unconvincing. There should be more time points settled within 1 month.

(7) Why choose tears as the detection object and what are the advantages and disadvantages of this? These information should be illustrated in the Introduction.

(8) This work is similar with the previous works by authors (Talanta, 2015, 134:333-339), the detection method and the immobilized membrane(EC) are same. Please explain more on the innovation of this work?

Reviewer 2 Report

This manuscript by Duong et al developing the ratiometric fluorescent glucose with membrane immobilized with glucose oxidase are impressive. The sensitive is quote low. However, the application on the tears is enigmatic. It is the weak point of the work. However, the development of non-invasive detection on the glucose of tears is a spurt of progress. The references should be closer to date. In the draft, the newest reference is published in 2014, six years before. The knowledge and introduction about non-invasive measurement should be updated. The article entitled with ‘Eyeglasses-based tear biosensing system: Non-invasive detection of alcohol, vitamins and glucose’ is suggested.

Typo: Line 114, 2.4 ……PS@C6PtP is PS@C6^PtP?

Reviewer 3 Report

Please find attached my comments

Reviewer 4 Report

The submitted manuscript reports the fabrication of a two-layer glucose-sensing membrane based on glucose oxidase (GOD) and oxygen sensing material. Three different supporting materials for GOD immobilization such as ethyl cellulose (EC), polyurethane hydrogel (D4) and a sol-gel mixture abbreviated as GA consisting  in aminopropyltrimethoxysilane and glycidoxypropyltrimethoxysilane were employed. Authors evaluate the sensing performance of the membranes in glucose solutions and artificial tears by fluorescence quenching method. The manuscript needs English language corrections. While the experimental part seems to be well structured with a lot of experimental data, the text used within the abstract, introduction and the one explaining the figures is rather difficult to understand/follow.

The manuscript needs MINOR revision. Please find below my comments:

  1. I recommend the authors to check/arrange/rewrite the text together with a native English speaker in order to express clear and concise their work.
  2. I suggest to integrate the part 1. Choice of materials from Results and discussion section to Introduction.
  3. The Introduction is not clear and must be revised. Please find below several examples:
  • Glucose biosensors can be classified into non-invasive and invasive measurement methods” (lines 42-43). Glucose biosensors cannot be classified into methods.
  • the advantage of fluorescent molecules for bio-sensing, such as extreme sensitivity”(lines 47-48). The “extreme sensitivity” should be defined/explained.
  • Concanavalin A (ConA) was immobilized on a high molecular weight fluorescein labeled dextran inside of a microdialysis fiber. The fluoroscein-labeled dextran was displaced from the binding site by glucose, causing an increase in fluorescence in proportion to the glucose concentration. This mechanism is developed by immobilizing ConA on a hydrogel to increase the linear response up to 600 mg/dL” (lines 50-54). It is not clearly expressed what the authors intend to point out.
  • Fluorescence is also coordinated with the enzyme GOD” (line 58). The verb “coordinated” is not properly used in this context.
  • in this work, ratiometric fluorescent glucose-sensing membranes were developed using a ratiometric fluorescent oxygen-sensing membrane.” (lines 62-64). The statement is unclear.
  1. I suggest to include a simple scheme to illustrate the glucose-sensing membranes.
  2. Include the units for Michaelis-Menten constant (mM)
  3. I strongly suggest to replace the word biocompatibility within the phrase “the biocompatibility of D4 polymer indicates that D4 can serve as a good supporting material for GOD” (lines 257-258). There is no biocompatibility test performed in the study.
  4. The scale (magnification) of the SEM imagines must be integrated in Fig 3.
  5. The abbreviation of the materials used for the study should be made only one time within manuscript.
  6. “1 mM to 10 mM” should be written 0.1 to 10 mM all over the manuscript (lines 26, 134, 141)
  7. Regarding the phrase “GOD immobilized on/in EC and D4 matrix preferred acidic medium (in the pH range of 5 to 7) to alkaline medium (pH 8 to pH 9)” (lines 390-391) I suggest, for instance, the following modifications:
  • there is no need to point out both the pH and the alkaline or acidic medium in the same sentence. Either you refer to the specific pH or you mention only the acidic/basic medium.
  • the phrase could be expressed different, for instance instead of “preferred” I would use “exhibited better sensing performance in” acidic media

Round 2

Reviewer 1 Report

I would like to recommend the publication of this revised version.

Author Response

Hi,

I have corrected the manuscript.

Best regards

JI Rhee

Reviewer 2 Report

This manuscript by Duong et al developing the ratiometric fluorescent glucose with membrane immobilized with glucose oxidase are impressive. The sensitive is quite low. However, the application on the tears is enigmatic. The development of non-invasive detection on the glucose of tears is a spurt of progress. The references are updated. However, the more references published within 5 years are welcomed.

Author Response

Hi,

We have upodated the references

Best regards

JI Rhee